# Systemic Anti-Cancer Therapy in Synovial Sarcoma: A Systematic Review

**DOI:** 10.3390/cancers10110417

**Published:** 2018-11-01

**Authors:** Richard F. Riedel, Robin L. Jones, Antoine Italiano, Chet Bohac, Juliette C. Thompson, Kerstin Mueller, Zaeem Khan, Seth M. Pollack, Brian A. Van Tine

**Affiliations:** 1Duke Cancer Institute, Duke University Health System, Durham, NC 27710, USA; richard.riedel@duke.edu; 2The Royal Marsden NHS Foundation Trust, Institute of Cancer Research, London SW3 6JJ, UK; robin.jones4@nhs.net; 3Department of Medical Oncology, Institute Bergonié, 33000 Bordeaux, France; A.Italiano@bordeaux.unicancer.fr; 4Immune Design Corporation, San Francisco, CA 94080, USA; 5ICON Epidemiology, ICON plc, Abingdon OX14 4RY, UK; juliette.thompson@iconplc.com; 6ICON Epidemiology, ICON plc, Vancouver, BC V6B 1P1, Canada; kerstin.mueller@iconplc.com (K.M.); zaeem.khan@iconplc.com (Z.K.); 7Fred Hutchinson Cancer Research Center, University of Washington, Seattle, WA 98109, USA; spollack@fhcrc.org; 8Division of Medical Oncology, Department of Medicine, Washington University School of Medicine, St Louis, MO 63110, USA; bvantine@DOM.wustl.edu; 9Siteman Cancer Center, St. Louis, MO 63110, USA

**Keywords:** synovial sarcoma, survival, chemotherapy, systemic anti-cancer therapy, systematic review

## Abstract

Synovial sarcoma (SS) is an aggressive malignancy which accounts for approximately 5–10% of all soft-tissue sarcomas. SS has pathologic and genomic characteristics that define it as a distinct subtype of soft tissue sarcoma (STS). STS subtypes continue to be recognized as distinct entities with specific characteristics, including differential chemo-sensitivity. The objective of this study was to conduct a descriptive review of current data on survival outcomes of systemic anti-cancer therapy specific to SS. A systematic literature review was conducted, using a custom search strategy to search EMBASE, Medline and CENTRAL for clinical trials and observational studies reporting overall survival (OS), progression-free survival (PFS) and/or response for cohorts of at least 50 SS patients. We identified 28 studies meeting these criteria, 25 of which were retrospective studies. Only three prospective studies were identified. Survival reports varied widely between studies based on the population, in particular on the disease stage, and reporting was heterogeneous in terms of the time points reported on. For patients with localized disease, reports of five-year PFS ranged from 26% to 80.7% and five-year OS from 40% to 90.7%, whereas five-year OS for patients with metastatic disease was very low at around 10%; and in one case, 0% was reported. Only four of the included publications reported outcomes by type of systemic anti-cancer therapy received. Our study draws attention to the fact that additional prospective studies to better define the most appropriate treatment for SS in all stages and lines of therapy are still needed.

## 1. Introduction 

There is limited differentiation in treatment guidelines between soft tissue sarcoma (STS) subtypes [1,2]. However, distinct subtypes with specific characteristics are being recognized; features that distinguish subtypes include morphology, immunohistochemistry, genomics, molecular profile and clinical features [3]. Identifying histology-specific treatment is important where STS subtypes are being classified and treated based on molecular and genetic characteristics as well as differential natural history and chemosensitivity [3,4]. Currently, there is no systemic anti-cancer therapy approved specifically for synovial sarcoma (SS). 

SS is an aggressive malignancy which accounts for approximately 5–10% of all STS [5,6,7,8]. The estimated incidence of SS is 1.5 per 1,000,000 in the US and 1.4 per 1,000,000 in the UK [7,9,10,11]. It is the most common non-rhabdomyosarcoma STS in children and young adults [8]. Approximately one-third of SS occurs in childhood, but peak incidence is in the third decade of life [12]. SS frequently arises in the extremities, but can also develop from serosal surfaces of almost any part of the body such as the pleura or pericardium [7,8].

SS has unique pathologic and genomic characteristics including histologic subtypes, biphasic and monophasic, and a characteristic translocation, t(X;18)(p11.2;q11.2) [13]. Other genomic characteristics such as the B-cell lymphoma 6 co-repressor (BCOR) upregulation and the immunoreaction of the nuclear tumor suppressor gene SMRRCB1/INI1 (INI1) were not found to be associated with a worse prognosis [14,15]. SS is one of the most common STS subtypes to express *NY-ESO-*1; greater than 80% of synovial tumors express *NY-ESO-1*, which is associated with worse prognosis [16,17,18,19,20].

There are no approved systemic therapies that target these histologic or genomic characteristics. Surgical resection is the standard treatment for localized SS, with consideration for use of neoadjuvant/adjuvant radiation and/or systemic anti-cancer therapy [7,21]. Prognosis for non-metastasized patients is often favorable for tumors <5 cm resected with adequate margins [22]; local recurrence occurs in approximately 17% of patients [14]. Metastatic disease is present in approximately 24% of the patients at diagnosis [14]. SS is associated with local recurrence and distant metastases. Metastases occur in 50–70% of cases [23]; most metastases develop in the lungs (80%), followed by bone (10%) and liver (5%) [5]. Locally advanced or metastatic SS tends to be treated initially with systemic anti-cancer therapy [24], however, there is no consensus regarding the most appropriate therapy regimen or sequence. The prognosis of patients with metastatic SS is limited regardless of treatment [25]. 

Treatment decisions for SS are informed by clinical features and published data obtained from trials that enrolled multiple STS subtypes. These trials were not designed to evaluate the impact of systemic anti-cancer therapy on a specific histologic subtype. This makes it difficult to evaluate the potential benefit for a specific histology such as SS.

No systematic overview of survival outcomes achieved with systemic anti-cancer therapy across studies specific to SS currently exists. The objective was to conduct a systematic literature review (SLR) of current published data on survival outcomes and responses to systemic anti-cancer therapy in SS. This review can serve to support development of future research and inform guidelines specific to SS using currently available anti-cancer therapies.

## 2. Materials and Methods

Methodology for the SLR was consistent with recommendations published in the Preferred Reporting Items for Systematic Reviews and Meta-Analyses (PRISMA) statement and by the Centre for Reviews and Dissemination [26,27].

### 2.1. Search Strategy and Study Selection

The search for peer-reviewed literature was limited to studies published in 2000 or later to evaluate the effect of systemic anti-cancer therapy on SS in the modern era. The following electronic databases were searched via the OVID search portal: Embase, MEDLINE, and the Cochrane Central Register of Controlled Trials (2000 to 30 January 2018). Search strategies are provided in the Appendix A. In addition, conference proceedings were searched for 2015–2017 for the following conferences: American Society of Clinical Oncology (ASCO), European Society of Medical Oncology (ESMO), and Connective Tissue Oncology Society (CTOS) Annual Meeting.

All abstracts and full text articles were reviewed according to the pre-specified eligibility criteria, outlined in Table 1, by two independent reviewers (Reviewers: JT, KM, ZK). Any studies where the reviewers disagreed on inclusion at either selection stage were referred to a third independent reviewer for a final decision. A PRISMA flow diagram (Figure 1) was developed indicating the numbers of studies included and excluded at each stage of the review. 

### 2.2. Data Extraction

Data were captured in an Excel^®^ (Microsoft) extraction template (included in the Appendix A) by a single reviewer and were independently validated by a second reviewer (Reviewers: Juliette Thompson, Kerstin Mueller, Zaeem Khan). Extracted data included study characteristics (study design, follow-up periods, and main inclusion/exclusion criteria), patient characteristics (mean age, sex distribution, prior therapies, disease stage, performance status, histology, and tumor size), treatment details, and efficacy endpoints (overall survival, progression-free survival, tumor response).

### 2.3. Data Analysis

Data were synthesized in tabular format; efficacy outcomes were stratified by age, histology, treatments, or by localized or advanced/metastatic disease when available in the publication. Overall survival (OS) and progression-free survival (PFS) were the primary outcomes of interest.

## 3. Results

A total of 3198 abstracts were identified in the search, and 28 met the inclusion criteria for this review (Figure 1). Of the included publications, 25 were retrospective studies reporting between 51 and 1318 SS patients, including a retrospective analysis of data from 15 clinical trials (9 Phase II, 2 Phase II/III and 4 Phase III) published between 1987 and 2015 [28], and three were prospective studies (two trials [1 phase III, 1 undefined] and one observational study) reporting between 70 and 188 SS patients (Table 2). The objectives of the included publications are listed in the Appendix A.

Despite the majority of publications being retrospective studies, most indicated SS was histologically or pathologically confirmed in the study sample. Only four studies indicated whether immunohistochemical results were required or recommended for diagnosis [12,29,30,31]. Only one study indicated classification of STS based on ICD-codes [32] only; the risk of misclassification was noted [33]. One publication discussed how histological review altered diagnosis for patients, 2/94 patients had been misdiagnosed [12]. Details of SS diagnosis was not provided in four studies [28,34,35,36].

Sixteen of 28 studies focused on localized disease [4,12,23,31,34,35,36,37,38,39,40,41,42,43], 10/28 studies included both localized and metastatic disease [11,25,28,29,30,44,45,46,47,48], and 2/28 studies focused on metastatic disease [49,50] (Table 1). Definitions of disease status were infrequently reported; generally just stating whether the population included was either localized or non-metastatic without staging information. Six studies reported disease stage at the start of the study. Details on whether patients were considered resectable were reported in 14 studies (Table 3). Of the included studies, eight were conducted in North America [4,23,33,34,39,42,45,48], 15 in Europe [11,25,28,29,31,35,36,37,38,40,41,43,46,47,49,50], two in Japan [30,44], and two were multi-regional [12,51]. Length of study follow-up ranged from 12.3 months [51] to 11.4 years (137 months) [38] for localized disease and from 51 months [44] to 78 months [45] for metastatic disease (Table 2).

Baseline characteristics of patients were heterogeneous (Table 3): median age ranged from 12 [36,41] to 42.5 years [29] and 26.7% [34] to 63% were female [42]. Monophasic tumors accounted for 18% [31] to 85% [42] and biphasic tumors for 15% [42] to 77% [31] of tumors. Primary tumors in the upper extremities ranged from 13% [34] to 37% [4] and in the lower extremities from 33% [34] to 85% [4].

Radiotherapy was received by 12% to 97% of patients in the localized setting [4,36], only one study reported radiotherapy use in metastatic patients (58%) [49]. Where reported, radiotherapy was received at a median dose of 45–65 Gy. [34,36,39,41,47]. Those receiving systemic anti-cancer therapy ranged from 14% for adult patients with localized SS [34] to 100% in the metastatic setting [49]. In the localized disease setting, between 51% [41] and 100% [34,35,37,46] of patients underwent surgery (Table 3).

Our study focused on outcomes with the use of systemic anti-cancer therapy. Only four of the included publications (Vlenterie [28], Ferrari [47], Gronchi [51], Sanfillipo [49]) reported outcomes by the type of systemic anti-cancer therapy. Other publications simply reported the proportion of patients receiving systemic anti-cancer therapy for which outcomes were reported.

PFS and OS outcomes were assessed using various definitions, imaging methods and follow-up periods at varying time points between the individual studies (Table 4). Among 11 studies reporting PFS, the endpoint was defined in eight studies as the time between diagnosis, study registration, or randomization and the latest date of event-free follow-up, disease progression, recurrence, or death. No definition of PFS was provided in three studies. OS was reported by 20 studies, of which only nine provided a definition as the time between diagnosis, study registration, randomization, or treatment initiation and the reported death date or last known date of follow-up. Most publications reported proportions of surviving patients at specific time points. Only six publications reported median follow-up durations for survival outcomes.

### 3.1. Localized Disease

Two studies reported treatment responses in patients with localized disease, one of which was response to surgery rather than systemic anti-cancer therapy, so was not considered relevant [36]. Al-Hussaini reported that one of 10 adult patients achieved a partial response and in the pediatric population two of 5 patients achieved response using the RECIST criteria (one partial, one complete) [34]. 

Among the publications reporting survival data for patients with localized disease, three-year PFS rates were reported by Ferrari (2015), ranging from 91.7% to 91.2% [12]. Three-year OS among the same patient subgroups were both 100%. Reports of five-year PFS ranged from 26% [37] to 80.7% [12] and five-year OS from 40% [37] to 90.7% [12]. Ten-year PFS was reported by two studies and ranged between 53% [39,47] and 89% [41]. Ten-year OS was reported by six studies and ranged from 51% [45] to 78% [41] (Table 4). 

Generally in studies of localized disease, patient outcomes were only reported for the whole study cohort and by disease stage at diagnosis (Table 4). Typically, grade 3 disease was associated with a shorter PFS and OS compared to grade 1 or 2 disease. Deshmukh reported a five-year PFS of 70% for grade 1 or 2 disease versus 55% for grade 3 disease [45]. Similarly, Kreig reported a five-year OS of 97% for grade 2 patients compared to 15% for grade 3 patients [38]. Vining observed a significant risk of mortality in patients with grade 3 disease compared to grade 1 or 2 (hazard ratio: 2.32, 95% confidence interval: 1.46–3.67; *p* < 0.001) [33]. In addition to the overall estimates of PFS reported (81.9% at 3 years), Ferrari provided results for patients in low (IRS group I, ≤5 cm tumor: 91.7% at 3 years) and intermediate (IRS group I, >5 cm tumor and all IRS group II: 91.2% at 3 years) risk groups [12]. Similarly, Al-Hussaini provided results for the adult (five-year PFS: 68.3%) and pediatric (five-year PFS: 74.9%) populations separately as well as combined (five-year PFS: 69.3%) [34]. 

Two retrospective studies reported OS for localized disease. Median OS in patients that underwent resection/amputation was 40.1 months (3.3 years) for patients treated between 1955 and 1999, all of which were classified as FNCLCC grade 3 [31]. Median OS in Italiano was 136 months (11.3 years) for patients with completely resected localized SS; 47.2% of these patients were classified as FNCLCC grade 3 [35]. 

### 3.2. Mixed Population (Locally Advanced and Metastatic Disease)

Four studies of mixed populations reported response to treatment, which was generally a combination doxorubicin (Dox) + Iifosfamide (Ifo). In Ferrari, overall response was 56% in patients receiving Dox or epirubicin+Ifo compared to 31% in patients treated with “other” chemotherapy as any line of treatment [47]. In Setsu, 20% of patients treated with primarily Ifo-based adjuvant or second-line therapy achieved a partial response [44]. 

Spurrell examined responses using radiology reports on patients treated from 1978–2003. Response to chemotherapy was recorded via retrospective viewing of radiology reports (most reports using Response Evaluation Criteria In Solid Tumors [RECIST], but historical reports using World Health Organization [WHO] criteria). They found that out of 92 patients that received first-line chemotherapy for advanced disease (metastatic, local recurrence not amenable to complete excision or primary tumor not amenable to excision), 38 had a response (Dox + Ifo: *n* = 18/30; Dox: *n* = 6/25; Ifo: *n* = 4/25). 

However, the details for those responses are not provided and 35 of the 92 patients had metastectomies which may have confounded response assessments [25]. Median OS for the total population in this study was 22 months [25].

Vlenterie reported a retrospective analysis of outcomes for cohorts of SS subjects enrolled across 15 prospective clinical trials in first-line systemic treatment for advanced STS published between 1987 and 2015. Treatment response using WHO or RECIST ranged between 21.5% and 33.3%; complete response was highest in CYVADIC (cyclophosphamide, vincristine, adriamycin, DTIC) treated patients (10%, *n* = 30), however, a similar complete response rate was observed for the larger cohort available for Dox + Ifo (*n* = 112, CR: 6.3%) [28]. The publication reported 27.8% of SS with treatment responses compared to 18.8% for other STS [28].

One-year PFS outcomes to first-line treatment were reported in locally advanced and/or metastatic disease by Vlenterie. One-year PFS was highest in patients who received CYVADIC (23.3%) [28]. The longest median PFS was 7.5 months in the Dox + Ifo treatment group. Greatest one-year OS outcomes were also achieved in the Dox + Ifo subgroup (66.3%) but the longest median duration for OS was 15.8 months in the CYVADIC group, reported in a trial published in 1995 [28]. 

The most consistently reported time point for PFS and OS in mixed population studies was at five years. Ferrari included patients with locally advanced (*n* = 40/271) or metastatic disease (*n* = 16/271) and reported a five-year PFS of 36.8% for all patients; ten-year PFS was 29.8% [47]. Five year OS across four studies ranged from 52% in an SS unspecified population reported by Corey [48] to 69.8% in a mixed population receiving surgery and chemotherapy consisting primarily of Dox + Ifo in Takenaka [30]. Deshmukh reported a ten-year OS of 42% in all patients in the cohort, which included localized (*n* = 99/135), locally recurrent (*n* = 9/135) and metastatic (*n* = 27/135) patients [45].

### 3.3. Metastatic Disease

Five retrospective studies reported treatment response in patients with metastatic disease. In Sanfilippo, a retrospective analysis of advanced SS patients in Italy, all patients (*n* = 61) treated with trabectedin assessed were evaluable for response according to the RECIST criteria; 15% achieved a partial response, and 35% had stable disease [49]. In a population where 89% had surgery in plus systemic anti-cancer therapy five-year OS was 10% [46], and in a population where 92.5% had surgery and 21.3% had radiotherapy in addition to systemic anti-cancer therapy five-year OS was 0% [30]. In a population where 90% received surgery and 71% received radiotherapy in addition to systemic anti-cancer therapy, 15% of patients were alive at ten years [45]. 

METASARC is a prospective observational analysis of SS (*n* = 188) (Table 5). It was not captured in the literature search due to the absence of a SS term in title or abstract, but included here due to its importance. The analysis consisted of metastatic patients, 80% of whom were treated with systemic anti-cancer therapy and 49% underwent a locoregional treatment of the metastasis: five-year OS was 7.14% [50], consistent with observations reported in retrospective studies in this literature review.

## 4. Discussion

The number of STS subtypes recognized as distinct entities is growing, including myxoid liposarcoma [62,63], undifferentiated pleomorphic sarcoma [4], and malignant peripheral nerve sheath tumor [64]. Subtype-specific treatments demonstrating superior outcomes include eribulin in liposarcoma [65], gemcitabine plus dacarbazine in leiomyosarcoma [66], and trabectedin in both liposarcoma and leiomyosarcoma [67,68]. Recently pembrolizumab showed responses in patients with undifferentiated pleomorphic sarcoma or dedifferentiated liposarcoma [69], also; both gemcitabine alone and gemcitabine-docetaxel appeared to be active in leiomyosarcoma [70]. 

SS is a distinct subtype needing a tailored approach for management and further research; the identification of a specific genetic alteration suggests targeted therapy could be of greater value than conventional therapy [15]. Due to a lack of data, no consensus guidelines exist regarding the optimal agents, sequencing or number of cycles of systemic anti-cancer therapy for SS. As a rare tumor type it lacks funding to conduct an adequately powered randomized study to address the question of which agents and sequencing are optimal to treat this disease. 

SS is frequently identified as a chemosensitive STS; in 1994 an original publication of 13 patients with recurrent or metastatic disease treated with high dose ifosfamide reported a response in all patients [71]. A retrospective review of advanced SS patients using RECIST found Dox + Ifo achieved responses at a higher rate compared to single agent doxorubicin or ifosfamide; however these results may have been confounded by the use of metastectomies [25]. There are no published prospective clinical trials that document SS as being more chemosensitive than the broad group of STS [72].

Few studies included linked specific anti-cancer therapy regimens with survival outcomes; most aggregated survival outcomes into a single estimate of efficacy for the studied population, rather than outcomes being reported independently by each agent or for the enrolled SS cohort. This makes it difficult to determine which anti-cancer therapy may be driving the survival benefit or outcomes for the SS cohort reported in the studies included in this review. 

A recent study by Wang calculated an aggregate five-year OS of localized and metastatic SS patients from the SEER registry for the years 2002 to 2013 to be 60.5% regardless of line of therapy [7]. Almost half of the 12 studies (5/12) we identified in this review that presented five-year OS for mixed populations of localized and advanced stage patients reported proportions of survivors that were within 5 percent points of this value (range: −8.5% to +11.5%). The SEER study also showed significantly shorter survival for patients with distant compared to localized stage SS, with median OS not reached among those with localized disease, and median OS with distant disease of under 20 months [7]. 

A limitation observed in the results of the SLR was the restricted number of prospective studies identified for inclusion. There were two reasons for this: first, the eligibility criteria of the review required that at least 50 SS patients were reported in a study. Secondly, the design of prospective studies in STS, which frequently recruit numerous sarcoma histologies, meaning that data for SS are available either only for a small cohort of patients (<50 SS participants), or that the data from the SS patients are incorporated in the overall STS results. In the conduct of this systematic review with a specific focus on SS, results reported in this way were not captured. Thus, the current available evidence can only be based on studies including all STS histological subtypes. As knowledge of the differences and molecular heterogeneity of the specific subtypes increases, the development of clinical trials is moving towards study designs for specific histologic subtypes. In current practice, data from prospective STS clinical trials are used to inform treatment practice for SS. 

Beyond the limitations of the review, the data reported in this SLR also have multiple limitations that should be considered when evaluating the reported results. They include but are not limited to the following: the majority of patients were treated prior to either RECIST v1.0 or v1.1 and the definition of responses are not always provided nor systematically applied; imaging methods were not consistent nor is it known if trained radiologists evaluated the response assessments; there is selection bias as to which systemic anti-cancer therapy was administered and the dose intensity is not reported; there was heterogeneity of patients treated in studies including locally advanced and metastatic disease; the majority of reported results are from retrospective or observational studies and as such there is no reporting of how data was censored or if data was source verified.

In order to mitigate the limitations which led to the paucity of data from the SLR, we evaluated data from trials which combined multiple histologies. From this we compiled prospective studies of systemic anti-cancer treatments that included a cohort of fewer than 50 SS patients as part of the STS enrolled population, and the key survival results they provided. Due to the focus on systemic treatment, the majority of these identified prospective studies (7/10) had recruited patients with metastatic disease. One study focused on locally advanced and metastatic [54] and two on relapsed or refractory disease [53,54]. 

The number of SS patients included in the prospective studies ranged from 7 to 38 (Table 5). Only two of the prospective studies reported outcomes beyond one year, which are compared to the outcomes presented in the studies identified in our SLR. Three-year PFS and OS was 56.3% and 81.3% for pediatric and adolescent patients treated with neoadjuvant vincristine, Dox + Ifo for locally advanced disease [54]. 

Across all relapsed/refractory and metastatic studies, median OS ranged from 6.7 to 13.4 months and PFS ranged from 1.0 to 5.6 months, respectively, in the maintenance study of placebo versus regorafenib arms of the REGOSARC trial [57]. 

Currently no standard approach exists for the use of systemic anti-cancer therapy in SS, neither are there studies that evaluate what sequencing of agents is recommended. An anthracycline plus ifosfamide combination is often considered in the first line treatment as it has a high response rate in STS [72]. However, the recent approval of olaratumab has led to the consideration of doxorubicin in combination with olaratumab as an alternate option in the anthracycline naive setting [73,74]. Treatment decisions tend to be made based on retrospective data and the treating physician and patient preferences to provide the best tumor outcomes and optimize the risk to benefit ratio. Clinical features including age, tumor size and location are also used in selecting an optimal regimen [14,33]. The use of adjuvant/neoadjuvant chemotherapy remains controversial as there are no definitive trials to support its use in practice, and a recent large multicenter histology driven use of neoadjuvant did not show benefit over standard systemic anti-cancer therapy [51]. However, factors associated with receipt of adjuvant chemotherapy were age <30, primary extremity site, grade 3 histology, tumor size >5cm and positive surgical margins [33].

## 5. Conclusions

SS represents 5–10% of all STS; as a result, published literature often does not focus on the specific histology, but rather reports on patients with SS as part of the larger STS group. There is a need to further understand the biology and develop novel therapeutics for this distinct subtype as well as to understand tumor genomics and mechanisms of resistance. Current clinical practice is to treat SS based on evidence from published prospective clinical trials that enrolled multiple subtypes of STS. This SLR has multiple limitations and is not generalizable; but gives an overview of the available data for the use of current systemic anti-cancer therapy options for treatment planning for patients with SS. Additional multi-center and global prospective observational studies and randomized clinical trials to better define the most appropriate treatment for SS in all stages and lines of therapy are still needed. Areas of future research cover a broad range of treatments including targeted therapies e.g., *NY-ESO-1*-directed therapies, tyrosine kinase inhibitors, Adoptive Cell therapies, epigenetics e.g., HDAC and EZH2 inhibitors, chromatin remodeling and metabolic pathways.

## Figures and Tables

**Figure 1 cancers-10-00417-f001:**
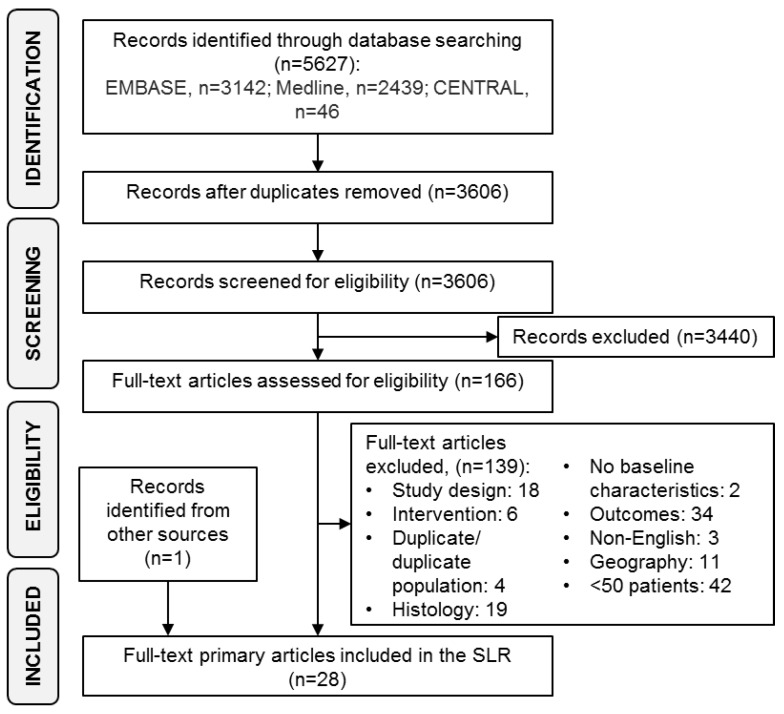
PRISMA diagram.

**Table 1 cancers-10-00417-t001:** Eligibility criteria.

**Patients**	Patients with reported SS (adult or pediatric)
**Interventions/Comparators**	Systemic anti-cancer therapy
**Outcomes**	OS, PFS, response (including overall, complete and partial response and stable and progressive disease), duration of response, time to next treatment
**Study Design**	Clinical trials (randomized or non-randomized) or observational study, excluding case reports and case series; minimum number of SS patients: 50
**Geography/Language**	Europe, North America, Japan, Australia or New Zealand; published in English
**Publication Year**	2000–2018

SS: Synovial Sarcoma, OS: Overall survival, PFS: Progression free survival.

**Table 2 cancers-10-00417-t002:** Included studies and their characteristics.

Author, Publication Year	Country	Number of Participants	Study Design	Follow-Up in Months Median (Range)
**Localized**				
de Silva, 2004 [31]	Scotland	51	Retrospective, Cohort	99
Scheer, 2016 [37]	Germany	52	Retrospective, Cohort	--
Krieg, 2011 [38]	Switzerland	62	Retrospective, Cohort	136.8 (3.6–331.2)
Beaino, 2016 [23]	USA	63	Retrospective, Cohort	85 * (13–210)
Orbach, 2011 [36]	Europe	88	Retrospective, Cohort	102 (3–168)
Shi, 2013 [39]	USA	92	Retrospective, Cohort	62.4
Eilber, 2007 [4]	USA	101	Retrospective, Cohort	58 (12–185)
Al-Hussaini, 2011 [34]	Canada	102	Retrospective, Cohort	67.2 (3.1–216)
Trassard, 2001 [40]	France	128	Retrospective, Cohort	128
Ferrari, 2015 [12]	Multi-national	138	Prospective trial	52.1 (13.8–104.4)
Brecht, 2006 [41]	Germany, Italy	150	Retrospective, Cohort	80 (6–250)
Italiano, 2009 [35]	France, Switzerland	237	Retrospective, Cohort	58 (1–321)
Canter, 2008 [42]	USA	255	Retrospective, Cohort	72 (0–287)
Vlenterie, 2015 [43]	Netherlands	461	Retrospective, Cohort	--
Vining, 2017 [33]	USA	544	Retrospective, Cohort	49.2
Gronchi, 2017 [51]	France, Italy, Poland, Spain	70 †	Prospective trial	12.3
**Locally advanced or metastatic**				
Takenaka, 2008 [30]	Japan	108	Retrospective, Cohort	54
Setsu, 2013 [44]	Japan	112	Retrospective, Cohort	51
Deshmukh, 2004 [45]	USA	135	Retrospective, Cohort	78 (20–420)
Guillou, 2004 [29]	France, Switzerland, Belgium	165	Retrospective, Cohort	37 (2–302)
Palmerini, 2009 [46]	Italy	250	Retrospective, Cohort	66
Ferrari, 2004 [47]	Italy	271	Retrospective, Cohort	65 (12–250)
Vlenterie, 2016 [28]	Europe	313	Retrospective, Cohort	--
Brennan, 2016 [11]	England	1318	Retrospective, Cohort	--
Corey, 2014 [48]	USA	3756	Retrospective, Database	--
Spurrell, 2005 [25]	UK	104	Retrospective, Cohort	--
**Metastatic disease**				
Sanfilippo, 2015 [49]	Italy, France, UK	61	Retrospective, Cohort	--
Savina, 2017 [50]	France	188 †	Prospective, Cohort	61 (1–300)

Abbreviations: ‘--’ = Not reported; * mean reported rather than median; † specifically SS patients.

**Table 3 cancers-10-00417-t003:** Population characteristics.

Author, Publication Year	Subgroup	N Baseline	Age (Years)	% Female	Treatments (%)			SS Histology (%)		Primary Tumor Site (%)			Resectable (%)
			Median (Range)		RT	Chemo	Surgery	Monophasic	Biphasic	Extremities	Upper Extremities	Lower Extremities	
**Localised**													
de Silva, 2004 [31]	All SS patients	51	33 (9–77)	49	25	27	100	18	77	92	--	75	100
Scheer, 2016 [37]	All SS patients, pediatric and adolescent	52	13.9 (0.9–20.9)	46	73	94	100	65	33	--	--	--	--
Krieg, 2011 [38]	All SS patients	62	-- (6–82)	58	--	--	--	56	44	76	--	--	--
Beaino, 2016 [23]	All SS patients	63	33 (4–74)	59	56	14	--	59	38		17	57	100
Orbach, 2011 [36]	All SS patients, pediatric and adolescent	88	12 (2–17)	47	--	93	74	--	--	--	20	46	--
	IRS I + Tumor size ≤ 5	17	--	--	12	65	88	--	--	--	--	--	--
	IRS I + Tumor size > 5	4	--	--	--	100	100	--	--	--	--	--	--
	IRS II + Tumor size ≤ 5	16	--	--	13	88	94	--	--	--	--	--	--
	IRS II + Tumor size > 5	12	--	--	33	92	92	--	--	--	--	--	--
	IRS III	39	--	--	56	97	79	--	--	--	--	--	--
Shi, 2013 [39]	All SS patients	92	35.3	50	--	23	--	--	--	--	24	58	100
Eilber, 2007 [4]	All SS patients	101	34 (16–75)	51	95	67	--	68	32	--	30	70	--
	IFO-based	68	33 (16–64)	54	97	100	--	68	32	--	37	63	--
	No chemo	33	38 (17–75)	46	91	0	--	70	30	--	15	85	--
Al-Hussaini, 2011 [34]	All SS patients	-	-	45	78	25	100	--	--	--	27	57	100
	Adult patients	87	37.6 (15–76)	48.3	83	14	100	--	--	--	30	60	100
	Pediatric patients	15	14 (0.4–18)	26.7	53	87	100	--	--	--	13	33	100
Trassard, 2001 [40]	All SS patients	128	33 (15–76)	58	80	57	99	57	35	76	--	--	99.2
Ferrari, 2015 [12]	All SS patients, pediatric and adolescent	138	13.7	44	--	--	--	--	--	--	--	--	--
	IRS I, tumor < 5 cm, no chemo	24	--	--	--	--	100	--	--	--	--	--	--
	IRS 1 tumor > 5 cm or IRSII, three to six courses of adjuvant chemo and RT	37	--	--	--	--	--	--	--	--	--	--	--
	IRS III or N1 tumor, six courses of chemo, delayed surgery, RT	77	--	--	--	--	--	--	--	--	--	--	--
Brecht, 2006 [41]	All SS patients, pediatric and adolescent	150	12 (1–21)	45	71	97	51	41	35	84	--	--	50.7
Italiano, 2009 [35]	All SS patients	237	35 (15–76)	51	--	--	100	--	--	--	21	65	100
Canter, 2008 [42]	All SS patients	255	34	50	--	--	--	66	34	--	23	58	--
	SYT-SSX1	73	41 (16–80)	36	66	47	--	56	44	--	20	59	--
	SYT-SSX2	59	35 (18–78)	63	75	40	--	85	15	--	19	56	--
Vlenterie, 2015 [43]	All SS patients	461	38 (2–89)	46	--	--	--	22	24	66	--	--	90.2
Vining, 2017 [33]	All SS patients	544	42 (29–55)	49.5	--	--	--	41.5	22.6	--	18.4	47.2	100
Gronchi, 2017* [51]	SS patients receiving standard chemotherapy	36	--	--	--	--	--	--	--	--	--	--	100
	SS patients receiving histotype-tailored therapy	34	--	--	--	--	--	--	--	--	--	--	100
**Mixed population (localized and metastatic)**													
Takenaka, 2008 [30]	All SS patients	108	37 (8–74)	59.3	21	77	93	63	30	67	--	--	96.3
Setsu, 2013 [44]	All SS patients	112	--	61	15	24	--	67	29	--	--	--	--
Deshmukh, 2004 [45]	All SS patients	135	31 (8–81)	37	71	38	90	--	--	--	16	65	--
Palmerini, 2009 [46]	Metastatic	46	40 (13–79)	39	--	--	89	74	22	--	--	--	--
	Localized	204	36 (7–83)	54	--	48	100	60	36	--	--	--	--
Ferrari, 2004 [47]	All SS patients	271	32 (5–87)	47	--	41	--	38	43		22	63	--
Vlenterie, 2016 [28]	All SS patients	313	40 (18–81)	39	41	100	43	--	--	56	14	41	--
Brennan, 2016 [11]	All SS patients	1318	--	48	--	--	71	--	--	65	--	--	--
	Adults	1136	--	48	--	--	70	--	--	66	--	--	--
	Children and adolescents	182	--	43	--	--	77	--	--	63	--	--	--
Corey, 2014 [48]	SS, biphasic	732	40	47	--	--	--	--	100	--	--	--	--
	SS, histology not specified	1820	41	48	--	--	--	--	--	--	--	--	--
	SS, spindle cell	1204	41	49	--	--	--	--	--	--	--	--	--
Spurrell, 2005 [25]	All SS patients	104	33	50	67	30	87	40	38	66	18	48	--
**Metastatic disease**													
Sanfilippo, 2015 [49]	All SS patients	61	37 (18–68)	58	--	100	--	--	--	57	--	--	0
Savina, 2017 * [50]	All SS patients	188	--	--	--	--	--	--	--	--	--	--	--

Abbreviations: ‘--’ = Not reported; IFO = Ifosfamide; IRS = Intergroup Rhabdomyosarcoma Staging; SS = Synovial sarcoma. * baseline characteristics only provided for total STS population.

**Table 4 cancers-10-00417-t004:** Survival outcomes.

Study ID	Population	N	PFS		OS	
			% PFS (95% CI)	Median, Months (95%CI)	% OS (95% CI)	Median, Months (95% CI)
**Localized**						
***Time point 1 year***						
Italiano, 2009 [35]	All SS patients	237	--	--	85 (82, 88)	136 (70, 204)
***Time point 3 years***						
Ferrari, 2015 [12]	Tumor site-axial	39	77.7 (60.2–88.2)	--	100 (-)	--
	Tumor site-extremities	99	83.8 (74.4–89.9)	--	96 (88.2–98.7)	--
***Time point 5 years***						
de Silva, 2004 [31]	All SS patients	51	--	--	56 (-)	40.1 (-)
Scheer, 2016 [37]	All SS patients	52	26 (-)	--	40 (-)	--
	Tumor site-axial	11	18.2 (-)	--	22.7 (-)	--
	Tumor site-non-axial	41	28.4 (-)	--	44.7 (-)	--
Krieg, 2011 [38]	All SS patients	62	--	--	74.2 (-)	--
	FNCLCC Grade 2	32	--	--	97 (-)	--
	FNCLCC Grade 3	11	--	--	18 (-)	--
Orbach, 2011 [36]	All SS patients, pediatric	88	68 (-)	--	85 (-)	--
Shi, 2013 [39]	All SS patients	92	56 (-)	--	61 (-)	--
	Chemo	21	67 (-)	--	65 (-)	--
	No Chemo	71	49 (-)	--	53 (-)	--
	Tumor site-extremities	75	57 (-)	--	60 (-)	--
	Tumor site-trunk	11	52 (-)	--	63 (-)	--
Al-Hussaini, 2011 [34]	All SS patients	102	69.3 (-)	--	80.3 (-)	--
	Adult	87	68.3 (-)	--	76.9 (-)	--
	Pediatric	15	74.9 (-)	--	100 (-)	--
	Chemo (all SS patients)	25	62.6 (-)	--	--	--
	No Chemo	77	71.5 (-)	--	--	--
Ferrari, 2015 [12]	All SS patients	138	80.7 (72.5, 86.7)	--	90.7 (82, 95.3)	--
Brecht, 2006 [41]	All SS patients	150	77 (-)	--	89 (-)	--
	Tumor site-extremities	129	79 (-)	--	90 (-)	--
	Tumor site-other (not extremities)	21	68 (-)	--	88 (-)	--
	Tumor status-T1	94	88 (-)	--	96 (-)	--
Italiano, 2009 [35]	All SS patients	237	--	--	64 (59, 69)	--
Vlenterie, 2015 [43]	All SS patients	461	--	--	63.5 (-)	--
Takenaka, 2008 [30]	Localized	91	--	--	81 (-)	--
Deshmukh, 2004 [45]	Localized (primary + locally recurrent)	108	--	--	69 (-)	--
Palmerini, 2009 [46]	Localized	204	58 (51, 66)	--	76 (69, 82)	--
***Time point 9 years***						
Italiano, 2009 [35]	All SS patients	237	--	--	46 (40, 52)	--
***Time point 10 years***						
de Silva, 2004 [31]	All SS patients	51	--	--	45 (-)	--
Krieg, 2011 [38]	All SS patients	62	--	--	61.2 (-)	--
	FNCLCC Grade 2	32	--	--	84 (-)	--
	FNCLCC Grade 3	11	--	--	0 (-)	--
Shi, 2013 [39]	All SS patients	92	53 (-)	--	56 (-)	--
	Chemo	21	67 (-)	--	65 (-)	--
	No Chemo	71	49 (-)	--	48 (-)	--
	Tumor site-extremities	75	55 (-)	--	55 (-)	--
	Tumor site-trunk	11	41 (-)	--	63 (-)	--
Brecht, 2006 [41]	All SS patients	150	89 (-)	--	78 (-)	--
Vlenterie, 2015 [43]	All SS patients	461	--	--	53.8 (-)	--
Deshmukh, 2004 [45]	All SS patients	108	--	--	51 (-)	
	Localized (primary + locally recurrent)	108	--	--	51 (-)	--
**Mixed population localized/locally advanced / metastatic**						
***Time point 2 years***						
Corey, 2014 [48]	SS, biphasic	732	--	--	85 (-)	--
	SS, NOS	1820	--	--	71 (-)	--
	SS, spindle cell	1204	--	--	77 (-)	--
***Time point 1 year***						
Vlenterie, 2016 [28]	All SS patients	313	18.7 (14.6, 23.3)	6.3 (5.9, 7.0)	63.7 (57.9, 68.8)	15 (13.9, 16.4)
	Anthracycline	121	17.5 (11.2, 24.9)	5.06 (4.3, 6.1)	62.6 (53.1, 70.8)	14.85 (12.2, 16.2)
	Dox + IFO	112	21.4 (14.4, 29.4)	7.47 (6.5, 8.7)	66.3 (56.7, 74.3)	14.98 (12.9, 18.9)
	CYVADIC	30	23.3 (10.3, 39.4)	6.08 (3.0, 10.8)	58.6 (38.7, 74.1)	15.8 (8.4, 23.1)
	IFO	42	15.5 (6.4, 28.5)	7.2 (5.9, 9.2)	65.6 (48.9, 78)	15.34 (11.7, 19.7)
	Other chemotherapy	8	0 (-)	2.27 (1.0, 9.0)	29.2 (1, 71.9)	10.45 (0.9, NR)
	Tumor site-extremity ^b^	174	16.1 (11–21.9)	6.21 (5.3–7.1)	63.1 (55.3–69.69)	15.5 (13.3–18.1)
	Tumor site-other ^c^	52	22.0 (11.9–34.2)	7.13 (5.1–8.9)	63.6 (48.6–75.3)	14.4 (11.6–18.7)
***Time point 3 years***						
Ferrari, 2015 [12]	IRS I, tumor <5 cm, no chemo	24	91.7 (70.6, 97.8)	--	100 (-)	--
	IRS 1 tumor >5 cm or IRSII, three to six courses of adjuvant chemo and RT	37	91.2 (75.1, 97.1)	--	100 (-)	--
	IRS III or N1 tumor, six courses of chemo, delayed surgery, RT	77	77.7 (60.2–88.2)	--	100 (-)	--
***Time point 5 years***						
Takenaka, 2008 [30]	All SS patients	108	--	--	69.8 (-)	--
	Tumor site-extremities	72	--	--	57.4 (-)	--
	Tumor site-trunk	36	--	--	78.1 (-)	--
Setsu, 2013 [44]	All SS patients	65	--	24 (-)	62 (-)	--
	Tumor site-distal extremities	39	48.3 (-)	--	79.8 (-)	--
	Tumor site-proximal extremities	63	36.5 (-)	--	50.9 (-)	--
Deshmukh, 2004 [45]	All SS patients	135	--	--	61 (-)	--
Palmerini, 2009 [46]	All SS patients	-	--	--	68 (-)	--
	Tumor site-lower extremities	-	59 (-)	--	75 (-)	--
	Tumor site-upper extremities	-	76 (-)	--	78 (-)	--
Ferrari, 2004 [47]	All SS patients	271	36.8 (-)	--	64.3 (-)	--
	Adjuvant chemo	61	55.3 (-)	--	71.5 (-)	--
	No adjuvant chemo	154	35.2 (-)	--	70.3 (-)	--
	IFO + Dox/Epi	-	52.3 (-)	--	--	--
	Other chemo	-	59.4 (-)	--	--	--
	Locally advanced disease ^a^	40	31.9 (-)	--	49.6 (-)	--
Corey, 2014 [48]	SS, biphasic	732	--	--	65 (-)	--
	SS, histology not specified	1820	--	--	52 (-)	--
	SS, spindle cell	1204	--	--	56 (-)	--
***Time point 10 years***						
Deshmukh, 2004 [45]	All SS patients	135	--	--	42 (-)	--
	Primary tumor	99	--	--	55 (-)	--
	Local recurrence	9	--	--	11 (-)	--
Ferrari, 2004 [47]	All SS patients	271	29.8 (-)	--	--	--
***Time point not reported***						
Spurrell, 2005 [25]	All SS patients	104	--	--	--	22 (-)
**Metastatic disease**						
***Time point 0.5 years***						
Sanfilippo, 2015 [49]	All SS patients	61	23 (-)	3 (-)	--	--
***Time point 5 years***						
Takenaka, 2008 [30]	Metastatic	17	--	--	0 (-)	
Palmerini, 2009 [46]	Metastatic	46	--	--	10 (-)	--
Brecht, 2006 [41]	Tumor status-T2	53	60 (-)	--	78 (-)	--
Savina 2017 [50]	Metastatic	188	7.14 (-)	19.7	--	--
***Time point 10 years***						
Deshmukh, 2004 [45]	Metastatic	27	--	--	15 (-)	--

Abbreviations: ‘--’ = Not reported; CYVADIC = Cyclophosphamide, vincristine, adriamycin and DTIC; Dox + IFO = Doxorubicin and ifosfamide; Ifo + Dox/Epi = Ifosfamide and doxorubicin or epirubicin. ^a^ Considered unresectable at diagnosis; ^b^ Consisting of lower and upper extremities; ^c^ Consisting of trunk, head and neck, abdominal, thorax, gastrointestinal, skin, visceral gynecological, and other.

**Table 5 cancers-10-00417-t005:** SS specific PFS and OS data from prospective studies with a cohort <50 patients.

Study ID	Total Cohort (N)	Synovial Sarcoma Cohort (N)	Treatment Arm	Progression Free Survival (PFS)					Overall Survival (OS)	
				Median, Months (95% CI)	Comparator vs. Reference for	HR (95% CI)	Time, Months	Rate, % (95% CI)	Median, Months (95% CI)	Rate, % (95% CI)
**Relapse/refractory**										
Pappo, 2014 [52]	163	23	R1507	1.3 (1.2, 1.3)	--	--	--	--	--	17
Sleijfer, 2011 (EORTC Study 62043) [53]	142	38	Pazopanib	5.3 (2.6, 6.3)	--	--	--	--	10.2 (7.5, 13.3)	NR
**Locally advanced or metastatic**										
Pappo, 2005 [54]	43	16	Neoadjuvant vincristine, ifosfamide, and doxorubicin ^a^	--	--	--	36	56.3 (SD: 12)	--	81.3 (SD:10)
**Metastatic disease**										
Chugh, 2009 [55]	185	22	Imatinib	1.92 (1.92, 3.96)	--	--	--	--	--	--
Kawai, 2015 [56]	76	7	Trabectedin	NR	Trabectedin vs. BSC	0.14 (0.03, 0.68)	--	--	--	--
Mir, 2016 (REGOSARC) [57]	90	13	Regorafenib	5.6 (1.4, 11.6)	Regorafenib vs. placebo	0.10 (0.03, 0.35)	3	77 (42, 92)	13.4 (5.3, NR)	92
			Regorafenib	--	--	--	6	38 (14, 63)	--	77
			Regorafenib	--	--	--	9	38 (14, 63)	--	--
	92	14	Placebo	1.0 (0.8, 1.4)	--	--	3	0	6.7 (2.2, NR)	85
			Placebo	--	--	--	6	0	--	64
			Placebo	--	--	--	9	0	--	--
Ray—Coquard, 2008 [58]	48 ^b^	46	Gefitinib	1.4	--	--	--	--		
Schoffski, 2011 [59]	128	19	Eribulin mesylate	2.6 (2.3, 4.3)	--	--	3	21.1	--	71.1 (43.7, 86.8)
Schoffski, 2013 [60]	111	17	Cixutumumab	1.5 (1.3, 2.6)	--	--	3	21.4 (5.2, 44.8)	13.0 (5.1, 16.5)	94.1 (65, 99.1)
Van der Graaf, 2012 (PALETTE) [61]	369	30	Pazopanib	--	SS vs. other STS	0.82 (0.51, 1.32)	--	--	--	--

Abbreviations: ‘--’ = Not reported; BSC = Best supportive care; EORTC = European Organisation for Research and Treatment of Cancer; IFO + Epi = Ifosfamide and epirubicin; NR = Not reached; SS = Synovial sarcoma; STS = Soft-tissue sarcoma. ^a^ Given with granulocyte stimulating factor; ^b^ was solely a SS cohort, but only 46 evaluable for efficacy endpoint.

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
