# Peer review of "Systemic Anti-Cancer Therapy in Synovial Sarcoma: A Systematic Review"

_cancers, 2018, doi:10.3390/cancers10110417_

Round 1
Reviewer 1 Report
The authors Riedel et al provided an overview of the clinical studies focusing on the treatment options and clinical care management of synovial sarcoma. To obtain this goal the authors performed a systematic literature review screening more than 3 thousand abstracts of which 28 met the inclusion criteria selected for this work. The development of prospective studies and new therapeutic strategies are urgently needed to improve synovial sarcoma patient outcomes.
This is a well documented work and could represents a the starting point for further investigations.
The manuscript would benefit from the following:
1. Taking in consideration the differential diagnosis, the authors should mention if molecular cytogenetic analysis and immunohistochemistry staining data are provided or not in the analyzed studies (i.e. SS18-SSX fusion gene)
2. Representative images of monophasic and biphasic synovial sarcoma morphologies should be included in the manuscript
3. No data about the dosage of neo/adjuvant radiotherapy (i.e. 50 Gy) are mentioned
4. In the discussion section the authors should include the following studies: Current classification, treatment options, and new perspectives in the management of adipocytic sarcomas DOI: 10.2147/OTT.S112580, Update on the role of trabectedin in the treatment of intractable soft tissue sarcomas DOI: 10.2147/OTT.S127955, Olaratumab: PDGFR-α inhibition as a novel tool in the treatment of advanced soft tissue sarcomas DOI: 10.1016/j.critrevonc.2017.06.006, Epigenetic ConFUSION: SS18-SSX Fusion Rewires BAF Complex to Activate Bivalent Genes in Synovial Sarcoma DOI: 10.1016/j.ccell.2018.05.011.
Minor corrections are required before publication.
Author Response
We thank the reviewer for their time and comments, we provide the following responses per comment:
The information for this was limited, but we have added a sentence and references to clarify where this is available in the included literature.
As the focus is on the treatment for synovial sarcoma rather than the morphology of the disease we are not sure that this image is necessary in this context. However the reviewer has strong feelings regarding the inclusion of this image we can revise the manuscript again to include this.
The information for this was limited, but we have added a sentence and references to clarify where this is available in the included literature.
Thank you - the provided references have been added to relevant places in the introduction and discussion.
Reviewer 2 Report
Congratulations to the authors for this extensive review on systemic chemotherapy in synovial sarcoma. The authors well describe that few studies have been published explicitly describing the effect of chemotherapy in the soft tissue sarcoma-subtype synovial sarcoma.
1) There is one minor comment regarding grammar:
Page 3: "Other genomic characteristics include the B-cell lymphoma 6 co-repressor (BCOR) upregulation and the immunoreaction of the nuclear tumor suppressor gene SMRRCB1/INI1 (INI1) were not found to be..."
please change to: "Other genomic characteristics as the B-cell lymphoma 6 co-repressor (BCOR) upregulation and the immunoreaction of the nuclear tumor suppressor gene SMRRCB1/INI1 (INI1) were not found to be..."
2) One recommendation for the introduction and discussion:
The authors may stress even more the fact that the most recent large multicentre trial on the use of neoadjuvant chemotherapy in a histology-driven approach did not show a benefit over standard chemotherapy (Gronchi A et al., 2017. Lancet Oncology). The authors may mention that targeted therapy could be of greater value in this respect than conventional chemotherapy, considering the presence of specific genetic alterations in synovial sarcoma.
Author Response
We thank the reviewer for their time and comments, we provide responses for each comment below:
Thank you - we have updated the sentence as suggested.
Thank you - we have included both references in the relevant places as suggested.